# Influencing Factors and Symbiotic Mechanism of the Integration of Medical Care and Disease Prevention during the COVID-19 Pandemic: A Cross-Sectional Survey of Public Hospital Employees

**DOI:** 10.3390/ijerph20010241

**Published:** 2022-12-23

**Authors:** Zhen Wu, Huiyi Tian, Dongjian Xu, Jiaying Chen, Yaqi Hu, Xiaohe Wang, Siyu Zhou

**Affiliations:** 1School of Public Health, Hangzhou Normal University, Yuhangtang St., Yuhang, Hangzhou 311121, China; 2School of Public Administration, Hangzhou Normal University, Yuhangtang St., Yuhang, Hangzhou 311121, China

**Keywords:** integration, medical care and disease prevention, mediating effect, public health, symbiotic

## Abstract

Background: The COVID-19 outbreak has accelerated the huge difference between medical care and disease prevention in Chinese medical institutions. This study aimed to investigate the relationship between the symbiotic units, environments, models, and effects of the integration of medical care and disease prevention. Methods: This cross-sectional study involved 762 employees of public hospitals in 11 cities in Zhejiang Province by random stratified sampling. We analyzed the influence paths of elements in the mechanism of integration of medical care and disease prevention and the mediating effect of symbiotic models among symbiotic units, symbiotic environments, and effects on this integration. Results: The path coefficient of the symbiotic unit on the symbiosis model was 0.46 (*p* < 0.001), the path coefficient of the symbiotic environment on the symbiosis model was 0.52 (*p* < 0.001). The path coefficient of the symbiotic unit and the environment was 0.91 (*p* < 0.001). The symbiotic models exhibited a partial mediation effect between symbiotic units and the effect of this integration. Sobel test = 3.27, β = 0.152, and the mediating effect accounted for 34.6%. Conclusions: It is suggested that health policymakers and public hospital managers should provide sufficient symbiotic units, establish collaborative symbiotic models, and improve the effects of integration of medical care and disease prevention in public hospitals.

## 1. Introduction

Integration of medical care and disease prevention refers to a new model that combines the functions of medical care and disease prevention in medical institutions [1]. This model adjusts for the separation of clinical medicine, public health and emergency management functions of public hospitals in the past, and promotes the transformation of public hospitals from disease-centered to health-centered through measures of resource integration and efficiency improvement [2]. The integration of medical care and disease prevention is an important mechanism for responding to public health emergencies. However, there is a huge difference between medical care and disease prevention in Chinese medical institutions, and the phenomenon of emphasizing medical care and ignoring public health has existed for a long time [3]. Over the past four decades, the Chinese government has attached great importance to disease treatment in public hospitals but has neglected disease prevention and the construction of a public health system [4]. This has led to the lack of disease prevention and control capabilities in Chinese public hospitals. The resulting hidden dangers lead to nosocomial infection in the early stage of public health emergencies, which has become the main risk in hospital management [5].

In 2009, the Chinese government’s medical and health system reform changed from focusing on diseases to focusing on health, and the society’s emphasis on public health work has gradually increased. However, the focus of the reform is on primary health care and primary health emergency management. The reform ignores the integration of medical care and disease prevention in public hospitals, resulting in low efficiency of public health emergencies in the hospital [6]. Medical services can directly improve the economic income and reputation of public hospitals; therefore, the decision makers of public hospitals have chosen to pay attention to medical services and not to disease prevention. This has led to the status of the public health department in public hospitals being reduced, and the development of the public health department is seriously lacking in the clinical department [7]. External factors have also influenced the emphasis on disease prevention in public hospitals. The government’s application of finances primarily to the prevention and treatment of public health emergencies has reduced investment in public hospitals, leading to further cuts in disease prevention in public hospitals [8]. The government encouraged family doctors to provide public health services to residents in clinics, leading to a reduction in the function of disease prevention in public hospitals [9]. Frequent incidents of food hygiene poisoning in Zhejiang province have caused residents to hesitate about the effectiveness of disease prevention in public hospitals, leading to a decline in the importance of the public health sector [10].

The continuing spread of the COVID-19 epidemic has posed a huge threat to human health, economic development, and social stability. The Chinese government has adopted the “Integration of medical care and disease prevention policy” and “dynamic zero-Covid policy” to stop the spread of the virus and to protect the health of the population [11]. In recent years, research on the “Integration of medical care and disease prevention” has continued to increase. From the perspective of epidemic risk, some experts have pointed out that the lack of administrative management was the reason for the fragmentation of the public health service system and the poor integration of medical care and disease prevention. Strengthening investments in health management, health performance, health insurance, and health education was recommended [12,13,14]. Some experts believed that the Centers for Disease Control and Prevention (CDC) assumed the responsibility for public health, and that public hospitals only needed to undertake the diagnosis and treatment of patients during the COVID-19 epidemic [15,16]. A study suggested that strengthening government financial investment in public health departments can improve the effect of integration of medical care and disease prevention [17]. Some studies have also suggested that the integration of medical care and disease prevention is not the responsibility of public hospitals, and public hospitals cannot be required to implement the integration of medical care and disease prevention in the form of health services [18,19]. Further research reported that the integration of medical care and disease prevention combines the functions of disease treatment and public health, and it needs to rely on the collaborative management of clinical departments, public health departments, and administrative departments to improve the ability of public hospitals to integrate medical care and disease prevention [20,21]. Previous studies have expounded on the integration of medical care and disease prevention from the aspects of government behavior and environmental factors. However, few research studies have analyzed the mechanism of the integration of medical care and disease prevention in public hospitals and the relationship between various elements. 

The COVID-19 pandemic has led to the need for healthcare services to be transformed to adapt to the changing public health environment. This study uses symbiosis theory to analyze the linkages and pathways of action between healthcare and disease prevention in public hospitals from the perspective of the sustainable development of healthcare service operations, explaining the connotations and roles of symbiotic units, symbiotic environments and symbiotic models in public hospitals from the perspective of healthcare and disease prevention, and providing new management strategies for the sustainable development of hospital management.

## 2. Theoretical Hypothesis

This study used the symbiosis theory as the research theory. The “symbiosis theory” proposes that there is an interdependent symbiotic relationship between different elements, which provides a new perspective for analyzing complex phenomena [22]. In human cell research, the literature has explored the symbiotic mechanisms between viruses and cells, suggesting that the symbiosis between viruses and normal cells needs to be considered in the treatment process in order to tailor the treatment plan [23,24,25]. In the field of biology, symbiosis theory refers to the formation of an interdependent relationship between different organisms through symbiosis, parasitism and cooperation. Symbiosis theory emphasizes the need for multiple elements within a system to achieve overall progress through interdependence and facilitation in the long-term operation of modern society [26]. In the field of healthcare services, symbiosis theory has been used to study how digital technology and healthcare services can integrate and promote each other in order to promote the improvement of healthcare services [27]. The need for Chinese public hospitals to generate economic benefits in the process of health care mechanism reform has led to hospitals treating and charging for diseases as their primary goal, thus losing the function of disease prevention. The COVID-19 pandemic created a shortage of medical resources, and the government required public hospitals to take on both disease prevention and medical roles. Therefore, this study uses symbiosis theory to analyze how disease prevention and medical treatment can be integrated and mutually reinforced in order to enhance the capacity of public hospital healthcare services. Symbiotic elements consist of the symbiotic unit, symbiotic environment, and the symbiotic model [28]. The integration mechanism of medical care and disease prevention during the epidemic involved the responsibilities, internal environment, and institutional model of each department in public hospitals. The symbiosis theory can be used to explain the relationship between various elements within public hospitals. This study integrated symbiosis theory into public hospital management, and discussed symbiotic units (human, financial, and equipment resources), symbiotic environments (information environment, clinical environment, and supervision environment), and symbiotic models (management model and operation mechanism), and analyzed the influence path of the above symbiotic elements on the effect of the integration of medical care and disease prevention. This study makes the following hypotheses:

### 2.1. Symbiotic Unit and Effect of the Integration of Medical care and Disease Prevention

The symbiosis unit refers to the collection of public hospital resources, including human resources, financial resources, and information resources [29]. The development of symbiotic units may affect the effectiveness of the integration of medical care and disease prevention. By increasing investment in human, financial and equipment resources, the government and public hospitals can ensure the rational operation of human, financial and equipment resources for medical care and disease prevention, reduce the profit-seeking behavior of hospitals arising from insufficient investment, and thus balance the functions of medical care and disease prevention. Conversely, insufficient investment in the symbiosis unit may lead to a reduction in the effectiveness of the integration of medical care and disease prevention. In this study, health human resources, health personnel competency, multi-department setup, health emergency response capacity, and financial resources of public hospitals were used as measurement variables.

Therefore, we propose Hypothesis 1.

**Hypothesis** **1.**
*The symbiotic unit positively affects the effect of the integration of medical care and disease prevention.*


### 2.2. Symbiotic Model and Effect of the Integration of Medical care and Disease Prevention

The symbiosis model is the internal management model of the public hospital, which reflects the management and operational effect of the public hospital system. The symbiotic model includes the distribution and integration of the medical service model, the public health model, and the hospital management model [30]. The development of the symbiosis model may affect the effectiveness of the integration of medical care and disease prevention. By using digital technology to optimize human resource management mechanisms, performance management mechanisms, emergency response mechanisms and training mechanisms, public hospitals can improve the synergistic efficiency of various departments in carrying out medical care and disease prevention, reduce duplication of staff work, and then motivate medical staff to work, ultimately promoting the effectiveness of the integration of medical care and disease prevention. Conversely, an unreasonable symbiosis model may lead to a decrease in the effectiveness of the integration of medical care and disease prevention. This study used the human resource management mechanism, performance management mechanism, information management mechanism, emergency mechanism, and training mechanism of public hospitals as measurement variables.

Therefore, we propose Hypothesis 2.

**Hypothesis** **2.**
*The symbiotic model positively affects the effect of the integration of medical care and disease prevention.*


### 2.3. Symbiotic Environment and Effect of the Integration of Medical Care and Disease Prevention

The symbiotic environment refers to the policies, regulations, and social environment of public hospitals, which can have a positive or negative impact on the integration of health care and prevention in the public hospital system [31,32]. The development of a symbiotic environment may influence the effectiveness of the integration of medical care and disease prevention. The government encourages the participation of medical staff in medical care and disease prevention by establishing a favorable policy environment. By improving the working environment in public hospitals, the job satisfaction of medical staff is enhanced, thus promoting the integration of medical care and disease prevention. Conversely, deterioration of the symbiotic environment may lead to a decrease in the effectiveness of the integration of medical care and disease prevention. This study used the policy environment, communication environment, technology environment, and data environment as measurement variables.

Therefore, we propose Hypothesis 3.

**Hypothesis** **3.**
*The symbiotic environment positively affects the effect of the integration of medical care and disease prevention.*


### 2.4. The Mediating Role of the Symbiotic Model among Elements

According to the symbiosis theory, the symbiosis model is the embodiment of the management and operation of the integration of medical care and disease prevention, and is affected by the symbiotic unit and the symbiotic environment [33]. Different symbiosis models have different effects on the effect of the integration of medical care and disease prevention [34]. The symbiotic model has a mediating effect among the symbiotic unit, environment, and the effect of the integration of medical care and disease prevention. 

The symbiosis unit belongs to the internal subjects of the medical care and disease prevention function of public hospitals, and it can further enhance the integration effect by realizing the synergy of multiple subjects through the development of the symbiosis model. Therefore, we propose Hypothesis 4.

**Hypothesis** **4.**
*The symbiotic model has a mediating role between the symbiotic unit and the effect of the integration of medical care and disease prevention.*


The symbiotic environment belongs to the external environment of the medical care and disease prevention function of public hospitals, and it can be developed through the symbiotic model to achieve the optimal allocation of environmental resources. Therefore, we propose Hypothesis 5.

**Hypothesis** **5.**
*The symbiotic model has a mediating role between the symbiotic environment and the effect of the integration of medical care and disease prevention.*


The symbiotic unit influences the improvement of the symbiotic environment through its resource input. The symbiotic environment provides a good working environment for the development of the symbiotic unit. Therefore, we propose Hypothesis 6.

**Hypothesis** **6.**
*The symbiotic unit and the symbiotic environment influence each other.*


We established the integration of the medical care and disease prevention model in a public hospital system based on symbiosis theory (Figure 1).

## 3. Materials and Methods

### 3.1. Research Design

This study included employees of the general public hospitals in 11 cities in Zhejiang Province as the research objects and samples. The main reasons are as follows: First, the investment of public hospitals in Zhejiang Province ranks in the forefront of China. In the last five years, Zhejiang Province has maintained an annual growth rate of 10% in investment in public health in medical institutions, while encouraging a growing number of medical institutions, ranking among the highest in China [35]. Second, Zhejiang Province is the first province in China to implement the integration of medical care and disease prevention. Third, the construction of the public health system in Zhejiang Province is relatively mature.

The survey period for this study was from December 2021 to February 2022. According to the “2020 Zhejiang Provincial Health Personnel Statistical Yearbook,” the number of medical personnel in general public hospitals in Zhejiang Province was 238,000. Therefore, to achieve the 95% confidence level, 5% margin of error, and 50% response distribution, a minimum sample size of 662 employees was required. Considering a 20% dropout rate, we planned to include 792 employees.

The inclusion criteria of the survey respondents were: (1) full-time employees of the hospital; (2) worked in the hospital for more than 3 years; (3) needed to be doctors, nurses, or administrators; (4) participated in the integration of medical care and disease prevention in the hospital. The exclusion criteria were: (1) unfilled all questionnaire questions; (2) unable to answer questions accurately. Participants were required to sign an informed consent form.

We used a random stratified sampling method to determine the sample size for each city based on the distribution of the number of general public hospitals in each city in Zhejiang Province (Table 1). We selected two hospitals in each city to implement the survey. Considering the large number of general public hospitals in Hangzhou, we selected 4 hospitals in Hangzhou, making a total of 24 hospitals in Zhejiang Province for the survey. The survey was designed using the Questionnaire Star Online website, and the online questionnaire was sent to each hospital’s HRM department (see Appendix A). The questionnaires were distributed randomly to medical staff within each hospital by their HRM department according to the inclusion and exclusion criteria. Participants were required to sign an informed consent form (B1).

### 3.2. Measure

The four dimensions of symbiotic unit, environment, model, and the effect of the integration of medical care and disease prevention were taken as the questionnaire items. These items were scored using a 5-point Likert scale (1 = completely disagree, 5 = completely agree). The scores for the separate items were summed and divided by the total number of items. The formula was Score = $\overset{n}{\underset{i=n}{\sum }}\left(a+\beta \cdots +\delta \right)/n$. A pre-questionnaire survey of 50 public hospital employees was completed before the formal study began.

The dependent variable is the effect of the integration of medical care and disease prevention, which refers to the perception of public hospital employees on the integration of medical care and disease prevention, including awareness, acceptability, policy implementation, and cross-departmental cooperation.

Independent variables included acceptance of symbiotic units, environments, and models (Table 2). The items of the symbiotic unit included human resource, health personnel competency, multi-department setup, health emergency response capacity, and financial resource. The items of the symbiotic environment included policy environment, communication environment, technical environment, and data environment. The items of the symbiotic model included human resource mechanism, performance management, information mechanism, emergency mechanism and training mechanism.

Demographic characteristics (gender, age, educational background, job title, department, occupation, and years of work) were set as control variables.

### 3.3. Reliability and Validity Test

We used metrics that reflect internal consistency to measure data reliability. By importing all items into the analysis, Cronbach’s alpha coefficient was 0.977 (>0.8). We performed the Kaiser–Meyer–Olkin (KMO) and Bartlett’s sphericity test, the KMO value was 0.974 (>0.9), the Chi-square statistic of Bartlett’s sphericity test was 15343.021, and the Sig value was <0.05, which was suitable for factor analysis.

The reliability of each dimension was tested separately using internal consistency, adding the correction term for overall correlation (CITC), and removing less reliable observed variables. The higher the CITC value, the higher the discriminative power of the corresponding item. A CITC value of 0.3 was considered the minimum acceptable, a CITC value greater than 0.4 was acceptable, and a CITC value greater than 0.6 was reasonable. The results showed that all CITC values were greater than 0.6, and all evaluation results were acceptable (Table 3). The Cronbach’s alpha coefficient of each variable was above 0.6, indicating that the reliability of the questionnaire is appropriate. The standard factor of the observed variable was above 0.5, indicating that the questionnaire data had good structural validity.

### 3.4. Statistical Analysis

Descriptive analyses were used to describe the demographic characteristics of public hospital employees. Symbiotic units, environments, models, and effect of the integration of medical care and disease prevention scores were quantified and expressed as mean (M) ± standard deviation. One-way analysis of variance and Student’s t test were used to analyze the differences in scores of symbiotic units, models, environments, and effects of the integration of medical care and disease prevention for different demographic characteristics. The Pearson correlation coefficient was used to analyze the correlation between symbiotic units, models, environments, and effects. The strength of correlations was described as weak (|r| < 0.3), moderate (0.3 < |r| < 0.5), or strong (|r| > 0.70). Binary logistic regression analysis was used to analyze the influencing factors of symbiotic units, models, environments, and effects. Ages were divided into two ranges based on median values: <35  years and  ≥35  years. Educational background was divided into primary (associate college and undergraduate) and advanced (master’s and PhD). The working years were divided into short term (≤10 years) and long term (>10 years). The rapid development of public health in China has its roots in China’s “new health system reform” in 2009, and the impact of the health reform on public hospital employees can be more clearly distinguished by a 10-year distinction [36]. Job titles were divided into primary (attending physicians, resident physicians, nurses, and general managers) and senior (chief physicians, deputy chief physicians, senior nurses, and senior managers). Occupations were divided into doctors, nurses, and managers. Departments were divided into clinical departments, public health departments, and administrative departments. To assess the scores of symbiotic units, environments, models, and effects of the integration of medical care and disease prevention, ≥3 point was considered a high score, and <3 point was considered a low score. A structural equation model was used to analyze the influence paths of symbiotic elements in the mechanism of the integration of medical care and disease prevention. The bootstrap method was used to analyze the mediating effect of symbiotic models among symbiotic units, environments, and effects of the integration of medical care and disease prevention. The advantages of using the bootstrap test method for the mediation effect test was that it allowed the variable to contain measurement errors and it included all the data. Standardized estimates and standard error results were obtained using the bootstrap program, and samples were estimated with 95% confidence intervals and by the maximum likelihood estimation method. The analyses were conducted using Python (version 3.9.0, Python Software Foundation, Beaverton, OR, USA) and IBM SPSS Amos (version 24.0, IBM Software Inc., New York, NY, USA)

## 4. Results

### 4.1. Participants and Demographics

A total of 792 questionnaires were distributed, and 780 questionnaires were returned. After excluding 18 invalid questionnaires, 762 valid questionnaires remained, with an effective response rate of 96.2%. The average age of the participants was 34.9 ± 6.5 years, and the average working years were 9.9 ± 7.6 years. The 31–40 age group constituted the largest number of public hospital employees, accounting for 59.3%. The percentage of undergraduates was 77.1%. The proportion of participants in the clinical department was 76.6%. The proportion of nurses accounted for 55.1%. The percentage of participants that have worked for less than 10 years was 66.4% (Table 4). Participants scored an average of 3.7 ± 0.9 points for symbiotic units, 3.8 ± 0.8 for symbiotic environments, 2.9 ± 0.9 for symbiotic models, and 3.3 ± 0.9 for effects of the integration of medical care and disease prevention.

### 4.2. Differences in Symbiotic Unit, Environment, Model, and Effect of the Integration of Medical Care and Disease Prevention

In terms of gender, there were significant differences in the symbiotic environment scores. Females scored higher on symbiotic environments than males. In terms of age, there were significant differences in the score of effect, and public hospital employees aged over 40 years had higher evaluations of the effect of the integration of medical care and disease prevention than employees under 40 years old. In terms of educational background, there were significant differences in the symbiotic environment scores. Public hospital employees with higher education (master’s and PhD degrees) scored lower on the symbiotic environment than employees with lower education (associate college and undergraduate). In terms of job titles, there were significant differences in the effect scores. Public hospital employees with senior titles had higher evaluations of the effect of the integration of medical care and disease prevention than employees with low primary titles. There were significant differences in the scores for their effect on the integration of medical care and disease prevention, symbiotic unit, and model among different departments. Both the public health department and the administrative department scored higher than the clinical department, with the administrative department scoring the highest. The symbiosis unit scores were significantly higher in both the public health department and administrative department than those in the clinical department. In terms of the symbiotic model, the administrative department scored the highest, while the public health department scored the lowest. In terms of occupation, there were significant differences in the scores of symbiotic unit, environment, model and effect of the integration of medical care and disease prevention. Both managers and doctors scored higher than nurses on the effect of the integration of medical care and disease prevention. Managers scored significantly higher than doctors and nurses in the symbiosis unit. The symbiotic environment scores of managers and nurses were significantly higher than those of doctors. Managers scored significantly higher than doctors and nurses in the symbiotic model. In terms of working years, there were significant differences in the symbiotic environment, model, and effect of the integration of medical care and disease prevention. The scores of public hospital employees with long working years (>10 years) were significantly higher than those of employees with short working years (≤10 years) (Table 5).

### 4.3. Influencing Factors of the Symbiotic Unit, Environment, Model, and Effect of the Integration of Medical Care and Disease Prevention

Descriptive analyses were used to describe the demographic characteristics of public hospital employees. Symbiotic units, environments, models, and effect of the integration of medical care and disease prevention scores were quantified and expressed as mean (M) ± standard deviation.

Department differences, occupation differences, and working years were the influencing factors of the symbiotic environment score. The public health department scored 1.68 times higher than the clinical department. The score of the administrative department is 1.45 times greater than the clinical department. Nurses scored 1.33 times higher than doctors. Employees with more than 10 years of working scored 1.67 times higher than those with less than 10 years of working.

Age, department, occupation, and working years were the factors that influenced the score of the symbiotic model. Employees over 35 years scored 1.33 times higher than those under 35 years. The public health department scored 3.86 times higher than the clinical department. Nurses scored 1.47 times higher than doctors. Managers scored 3.79 times higher than doctors. Employees with more than 10 years of working scored 1.53 times higher than those with less than 10 years of working.

Gender, age, job title, department, occupation, and working years were the influencing factors of the effect of the integration of medical care and disease prevention. Females scored 1.81 times higher than males. Employees over the age of 35 years scored 1.36 times higher than those under the age of 35 years. The senior title score was 1.98 times greater than the primary title. The public health department scored 1.52 times higher than the clinical department. The administrative department scored 1.65 times higher than the clinical department. Managers scored 1.63 times higher than doctors. Employees with more than 10 years of working scored 1.18 times more than those with less than 10 years of working (Table 6).

### 4.4. Pearson Correlation Coefficient Analysis of Symbiotic Elements

The effect of the integration of medical care and disease prevention, symbiotic unit, environment, and model are positively and strongly correlated with each other (Table 7). The correlation coefficient between the effect and the symbiotic environment was greater than 0.8. The correlation coefficients between symbiotic unit, environment, and model were greater than 0.8.

### 4.5. Path Analysis of the Symbiotic Unit, Environment, Model, and Effect of the Integration of Medical Care and Disease Prevention

The fit index parameters of the structural equation model were examined in terms of absolute fitness, value-added fitness, and parsimonious fitness. The Chi-square/df Ratio, was 4.737 (<5.6), Goodness of Fit Index (GFI) was 0.916 (>0.9), and Incremental Fit Index (IFI), and Comparative Fit Index (CFI) were 0.969 and 0.969, respectively, which were higher than 0.9. The Parsimony-Adjusted Measures Index (PNFI) was 0.810 (>0.5), Root Mean Square Error of Approximation (RMSEA) was 0.070 (<0.08), Tucker Lewis Index (TLI) was 0.963 (>0.9), and the comprehensive data showed that the correction parameters of the fitted indicators were in accordance with the standard reference values, indicating that the constructed structural equation model had a good overall fit with the data.

The structural equation model reflected the relationship between the latent variables (Figure 2). The results showed that the path coefficient of symbiotic unit on the effect of the integration of medical care and disease prevention was 0.29, the path coefficient of symbiotic model on effect of the integration of medical care and disease prevention was 0.33, and the path coefficient of symbiotic environment on effect of the integration of medical care and disease prevention was 0.29. The latent variables positively correlated with the effect, indicating that the better the symbiotic unit, environment, and model, the better the effect of the integration of medical care and disease prevention.

Table 8 shows the results of model testing, including path coefficients and corresponding levels of significance. The path coefficient of the symbiotic unit on the symbiosis model was 0.46, the path coefficient of the symbiotic environment on the symbiosis model was 0.52. The symbiotic unit and environment directly affected the symbiotic model. Unlike the one-way pathways of other latent variables, the symbiotic unit and environment were co-variant relationships. The path coefficient of the symbiotic unit and environment was 0.91. The symbiotic unit and environment are positively and strongly correlated. The symbiotic environment and symbiotic units interact with each other and behave as bidirectional pathways. 

### 4.6. Mediating Effect of Symbiotic Model

The results of the structural equation model showed that all pathways were significant, which verified the research hypotheses H1, H2, H3, and H6. Mediating effects in H4 and H5 were analyzed using the bootstrap mediation effect test. The bootstrap mediation effect results showed that the direct effect value of the symbiotic unit on the symbiotic model was 0.462, and the direct effect value of the symbiotic model on the effect of the integration of medical care and disease prevention was 0.328 (Table 9). After controlling for the symbiotic model, the indirect effect value of the symbiotic unit on the effect was 0.152, and the total effect value of the symbiotic unit on the effect was 0.439 (Figure 3). The significance of a,b,c values is *p* < 0.001, and the significance of c’ was less than 0.001; this suggested that the symbiotic model had a mediating effect. We used ab/c to calculate the value of the mediation effect. It was concluded that the mediation effect accounted for 34.6% of the overall effect, and the proportion of the direct effect to the overall effect was 65.4%, indicating that the symbiotic model played a partial mediating role between the symbiotic unit and the effect of the integration of medical care and disease prevention.

The results of the bootstrap mediating effect revealed that the mediating effect value of the symbiotic model between the symbiotic environment and effect was not significant. We failed to validate H5.

We made use of the Sobel test to improve the test validity and substituted the corresponding value into Equation (1):(1)$$\bar{Z}=ab/{\left(Sa^{2}b^{2}+Sb^{2}a^{2}\right)}^{\wedge }1/2$$

*a* = 0.462, *S_a_* = 0.054; *b* = 0.328, *S_b_* = 0.093, calculated $\bar{Z}$ = 3.27. After being checked by MacKinnon’s critical table, 3.27 was greater than 0.09 (*p* < 0.05), and the results showed that the mediating effect was significant. The results of the mediation effect test showed that the indirect effect value of the symbiotic model on the effect was 3.27 (Table 10). The mediating effect of symbiotic unit → symbiotic model → effect was significant. The results of this study validate H4.

## 5. Discussion

### 5.1. Influencing Factors of Symbiotic Unit, Environment, Model, and Effect of the Integration of Medical Care and Disease Prevention

Age was an influencing factor for symbiotic model and effect. Older public hospital employees had higher perceptions of institutional awareness and effects than younger employees. The integration of medical care and disease prevention was an emergency mode after the occurrence of public health events. Older employees in Chinese public hospitals, who mostly experienced two public health events, severe acute respiratory syndrome (SARS) and COVID-19, were more affected by such events versus younger employees, leading to higher perceptions of model and effect [37]. Related to this result, employees with longer working years had higher perceptions of the symbiotic environment, model, and effects. After a public health emergency in China, employees of public hospitals with higher qualifications were often required to directly participate in health emergency work; thus, their perception was higher than younger employees [38]. Public health department employees had higher perceptions of the symbiotic unit, environment, model, and the effect of the integration of medical care and disease prevention than clinical department employees and administrative employees. The main reason is that starting from 2021, the Zhejiang provincial government required that medical institutions above the second level must set up a public health department. The public health department received government funding in health resources and systematically trained public health department employees [39]. Since administrators were responsible for public health training in public hospitals, their knowledge of the symbiotic unit, environment, and models was also higher than clinicians [40]. However, employees of clinical departments have been engaged in medical technology work for a long time, and their work content does not include public health knowledge, which led to their lack of awareness of the integration of medical care and disease prevention [41]. It is important to increase awareness by conducting training on the integration of medical care and disease prevention for clinical department employees. Unlike previous studies that found that doctors were more aware of public health [42,43], this study found that nurses were more aware of symbiotic units than doctors. The reason was that in China, the occurrence of nosocomial infections had a greater impact on the careers of nurses, which made nurses more sensitive to health emergencies [44]. 

### 5.2. Symbiotic Unit Significantly Positively Affected the Effect of the Integration of Medical Care and Disease Prevention

Sufficient symbiotic unit elements contributed to improve the effect of the integration of medical care and disease prevention. The variable coefficients of the items B2: Health personnel competency, B3: Multi-department setup, and B4: Health emergency response capacity were relatively large, indicating that the symbiotic unit significantly enhances the effect of the integration of medical care and disease prevention through the coordination of human resources, multi-department, and response mechanism. More than half of the participants noted that the insufficient number of employees in the public health department restricted the effect of the integration of medical care and disease prevention. The reduced capacity of health personnel also led to a reduction in the effectiveness of disease treatment and prevention. Required competencies for successful chronic disease prevention and health promotion encompass leadership, epidemiology, program practice, and evaluation, among others [45]. However, the training of doctors and nurses in public hospitals lacks these components, leading to a decline in the capacity of health personnel. The lack of time for education and training in disease treatment and prevention due to increased working hours also contributed to this decline in capacity [46]. In addition, doctors and nurses were not able to perform at their normal capacity due to the increased work pressure during the COVID-19 pandemic [47]. Although it has experienced SARS and the COVID-19 epidemic, the main investment in China’s health emergency management is still in the CDC, and the investment in the public health department of public hospitals was less and had just begun [48]. In addition, the profit-making requirements of public hospitals have caused public hospitals to focus on the income of clinical departments, while ignoring the needs of public health departments [49]. During the study period, two waves of Omicron outbreak occurred in Zhejiang Province, and the clinical departments, public health departments, and administrative departments of public hospitals were required to cooperate in epidemic prevention. Healthcare facilities with a symbiotic model demonstrated higher treatment efficiency and faster response times in response to the COVID-19 pandemic. This has directly reduced the health risks to the population of Zhejiang Province. It was necessary to further strengthen financial investment to promote the balanced development of public health departments, clinical medical departments, and administrative departments [50].

### 5.3. Symbiotic Environment Significantly Positively Affected the Effect of the Integration of Medical Care and Disease Prevention

The items C1: Policy environment, C2: Communication environment, and C3: Technical environment had relatively large coefficients of observed variables, which indicated that a comprehensive symbiotic environment was the main factor to promote the effect of the integration of medical care and disease prevention. With the Chinese government’s increasing emphasis on the integration of medical care and disease prevention and the changing role of public hospitals in the epidemic, the policy of the government administrative department has become the main source of public hospitals to promote the effect of the integration of medical care and disease prevention [51]. There has been a communication gap between clinical departments and public health departments because public health departments in public hospitals have been neglected in the past. This weakening of communication was the main reason for nosocomial infection in public hospitals during the epidemic [52]. Establishing a harmonious communication environment between multiple departments will help improve the effect of the integration of medical care and disease prevention. Through the sharing of epidemic and diagnosis information via digital technology, the established public hospital information environment facilitates multi-department cooperation [53]. In the state of health emergency response, digital technology can provide an early warning for public hospitals through data analysis, leading to a quicker response to the epidemic within the hospital [54].

### 5.4. Symbiotic Model Significantly Positively Affected the Effect of the Integration of Medical Care and Disease Prevention

Items D2: Performance management, D3: Information mechanism, and D5: Training mechanism had relatively large coefficients of observed variables, which indicated that the establishment and improvement of the symbiosis model was of great significance to the effect of the integration of medical care and disease prevention in public hospitals. Through the performance management of employees in public hospitals, it was helpful to encourage clinical employees to participate in the integration of medical care and disease prevention, thereby improving the effect of the integration of medical care and disease prevention [55]. The information-sharing model could break the shortcomings of the previous work of each department in the process of integration of medical care and disease prevention [56]. The perfection of the digital technology environment and the communication environment in the symbiotic environment provided the basis for the information-sharing model. The establishment of an information management system in public hospitals breaks down communication barriers and improves management efficiency, based on technical maturity such as ensuring a smooth network. Smart hospital systems can respond quickly to public health emergencies and coordinate the preparation of supplies across departments [57]. Digital storage of disease information improves the efficiency of medical staff and allows data to be quickly shared with relevant departments for early warning [58]. After the information-sharing mechanism was established, various departments could achieve barrier-free communication and avoid nosocomial infection [59]. In the past, the training content of Chinese public hospitals was medical technology, and the content of medical and preventive integration was not included in the training [60]. With the outbreak of COVID-19, the public health department has become an equally important department when compared to the clinical department. Recent training not only incorporated the medical knowledge of COVID-19, but also incorporated management measures of the integration of medical care and disease prevention in public hospitals [61]. The symbiosis model improved the effect of the integration of medical care and disease prevention by institutionalizing management measures.

### 5.5. Mediating Role of the Symbiotic Model between the Symbiotic Unit and the Effect of the Integration of Medical Care and Disease Prevention

The symbiotic model had a partial mediating effect between the symbiotic unit and the effect of the integration of medical care and disease prevention, with a mediating effect value of 0.439, accounting for 34.6% of the total effect. This indicated that the symbiotic unit promoted the optimization of the symbiotic model and improved the effect through the mediating effect of the symbiotic model. For each additional symbiotic unit, the effect of the integration of medical care and disease prevention can be increased by 0.439. Through the resource allocation of health human resources and health financial resources of the symbiotic unit, the systematic optimal allocation of health resources in public hospitals can be achieved, thereby improving the management efficiency of public hospitals [62]. The realization of the effect of the integration of medical care and disease prevention needs to rely on the institutionalization of symbiotic units and models. This study believes that it is beneficial to respond to the risk of public health events by changing from “relying on people for management” to “relying on institutionalized management.” Due to the increase in the government’s investment in the integration of medical care and disease prevention, the policy environment directly affected the symbiotic effect, resulting in the symbiotic environment not serving as an intermediary variable [63]. Public hospitals need to adjust the health policy according to the characteristics of the hospital and pay attention to the effect of the policy environment on the integration of medical care and disease prevention.

Some limitations of this study should be noted. This study only included participants who were employees of general hospitals; employees from specialized and primary hospitals were not included. Therefore, the conclusions of this study are only applicable to general hospitals. The concentration of a proportion of the medical staff working in the same hospital may have led to variability in the evaluation of the effectiveness of the integration of medical care and disease prevention in different hospitals in this study. As this study is a cross-sectional survey, it is not possible to determine the temporal causality of the effects produced by the development of symbiotic units, symbiotic environments and symbiotic models. We plan to further optimize the representativeness of the sample selection in a subsequent study.

## 6. Conclusions

This study analyzed the system of integration of medical care and disease prevention of Chinese public hospitals and analyzed the mechanism of symbiotic unit, model, and environment on the effect of medical care and disease prevention. The symbiotic unit, environment, and mode all directly affected the effect. Among them, the symbiotic model had the greatest impact. During the ongoing COVID-19 period, public hospitals need to strengthen the institutionalization of the symbiosis unit, establish a harmonious symbiosis environment, and form a reasonable symbiosis model, thereby enhancing the effect. The symbiotic model partially mediates between the symbiotic unit and the effect. Through the investment of the symbiotic unit and the improvement of the multi-departmental collaboration system, the intermediary effect of the symbiotic model can be exerted, and the effect of the integration of medical care and disease prevention in public hospitals will be improved. Improving the public health capabilities of employees in clinical departments, public health departments, and administrative departments through training will also contribute to the integration of medical care and disease prevention in public hospitals.

## Figures and Tables

**Figure 1 ijerph-20-00241-f001:**
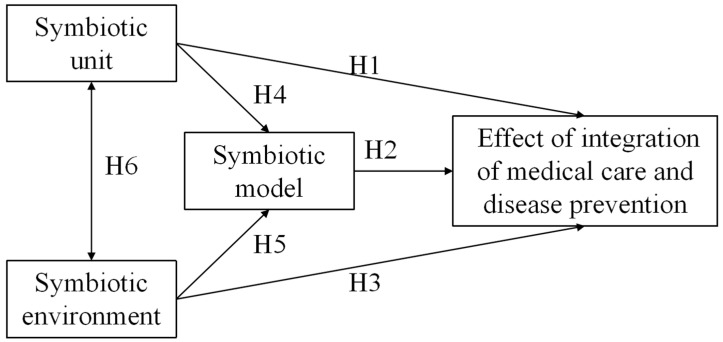
Integration of medical care and disease prevention model into a public hospital system.

**Figure 2 ijerph-20-00241-f002:**
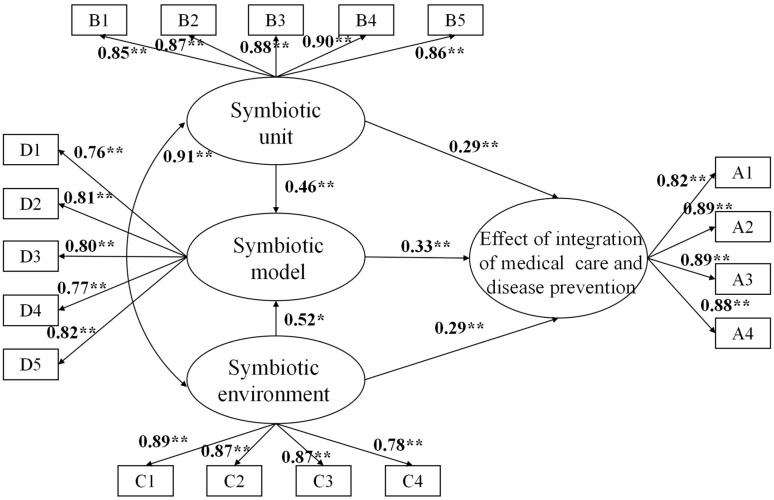
Model of the integration of medical care and disease prevention. Note: ** significant levels of 1%. * significant levels of 5%.

**Figure 3 ijerph-20-00241-f003:**
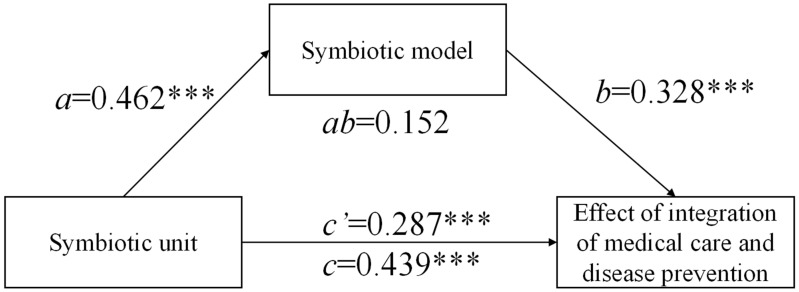
Mediating effects of symbiotic unit, model, and effect. Note: *** significant levels of 0.1%.

**Table 1 ijerph-20-00241-t001:** Survey sample size by cities in Zhejiang Province.

City	Number of General Hospitals	Survey Sample Size
Hangzhou	22	232
Ningbo	8	84
Wenzhou	10	106
Jiaxing	4	42
Huzhou	6	63
Shaoxing	7	74
Jinhua	5	53
Quzhou	3	32
Zhoushan	3	32
Taizhou	3	32
Lishui	4	42
Total	75	792

**Table 2 ijerph-20-00241-t002:** Latent variable and items.

Variable	Items	Content	Code
Effect of the integration of medical care and disease prevention	Awareness	Employees’ perception of the integration of medical care and disease prevention	A1
	Acceptability	Employees’ acceptance of the integration of medical care and disease prevention	A2
	Policy implementation	Employees’ implementation of integration policy of medical care and disease prevention	A3
	Cross-departmental cooperation	Cooperation between public hospital departments in the integration policy of medical care and disease prevention	A4
Symbiotic unit	Human resource	Status of human resources for health	B1
	Health personnel competency	Comprehensive quality of health personnel	B2
	Multi-department setup	Set up medical departments, public health departments	B3
	Health emergency response capacity	Health emergency management capabilities	B4
	Financial resource	Allocation of financial resources	B5
Symbiotic Environment	Policy environment	The popularization of the integration of medical and prevention policy	C1
	Communication environment	Employees’ cooperation platform	C2
	Technical environment	The support of digital information technology	C3
	Data environment	Database of the integration of medical care and disease prevention	C4
Symbiotic Model	Human resource mechanism	Human resource mechanism of the integration of medical care and disease prevention	D1
	Performance management	Performance appraisal mechanism	D2
	Information mechanism	Information sharing mechanism	D3
	Emergency mechanism	Setting of emergency management departments	D4
	Training mechanism	Regular training mechanism	D5

**Table 3 ijerph-20-00241-t003:** Reliability test of variable indicators.

Items	Code	CITC ^1^	Varimax Post-Rotation Factor Loadings			
			F1	F2	F3	F4
Effect (A)	Cronbach alpha = 0.925					
Awareness	A1	0.916	0.786			
Acceptability	A2	0.896	0.747			
Policy implementation	A3	0.897	0.738			
Cross-departmental cooperation	A4	0.897	0.729			
Symbiotic unit (B)	Cronbach alpha = 0.943					
Human resource	B1	0.934		0.749		
Health personnel competency	B2	0.931		0.716		
Multi-department setup	B3	0.928		0.699		
Health emergency response capacity	B4	0.928		0.634		
Financial resource	B5	0.925		0.563		
Symbiotic environment (C)	Cronbach alpha = 0.950					
Policy environment	C1	0.942			0.685	
Communication environment	C2	0.940			0.682	
Technical environment	C3	0.938			0.665	
Data environment	C4	0.937			0.629	
Symbiotic model (D)	Cronbach alpha = 0.913					
Human resource mechanism	D1	0.909				0.719
Performance management	D2	0.883				0.708
Information mechanism	D3	0.878				0.661
Emergency mechanism	D4	0.878				0.621
Training mechanism	D5	0.875				0.597

^1^ CITC: correction term for overall correlation.

**Table 4 ijerph-20-00241-t004:** Descriptive statistics of the participants (n = 762).

Variable	Items	Number	%
Gender	Male	193	25.3
	Female	569	74.7
Age	25–30	200	26.2
	31–40	452	59.3
	41–50	95	12.5
	51–60	15	2.0
Education	Associate college	23	3.0
	Undergraduate	587	77.1
	Master	136	17.8
	PhD	16	2.1
Job title	Primary	638	83.7
	Senior	124	16.3
Department	Clinical	584	76.6
	Public health	106	13.9
	Administration	72	9.5
Occupation	Doctor	270	35.4
	Nurse	420	55.1
	Manager	72	9.5
Working years	≤5	302	39.6
	6–10	204	26.8
	11–15	96	12.6
	>15	160	21.0

**Table 5 ijerph-20-00241-t005:** Univariate analysis of the symbiotic unit, environment, model, and effect of the integration of medical care and disease prevention.

Variables	Items	Unit	*p*	Environment	*p*	Model	*p*	Effect	*p*
Gender	Male	3.2 ± 1.0	0.58	3.6 ± 1.0	0.031	2.8 ± 1.0	0.202	3.0 ± 1.1	0.21
	Female	3.3 ± 0.9		3.8 ± 0.9		2.9 ± 0.8		2.8 ± 0.9	
Age	25–30	3.3 ± 0.9	0.75	3.7 ± 0.8	0.86	2.9 ± 0.8	0.23	2.8 ± 0.9	<0.01
	31–40	3.3 ± 1.0		3.7 ± 1.0		2.9 ± 1.0		2.8 ± 1.0	
	41–50	3.2 ± 0.6		3.6 ± 0.6		3.0 ± 0.6		3.1 ± 0.7	
	51–60	3.3 ± 0.5		3.8 ± 0.6		3.2 ± 0.4		2.9 ± 1.0	
Education	Associate college	3.1 ± 1.4	0.56	3.5 ± 1.2	0.022	2.6 ± 1.2	0.39	2.5 ± 1.4	0.12
	Undergraduate	3.3 ± 0.9		3.8 ± 0.8		2.9 ± 0.9		2.9 ± 1.0	
	Master	3.2 ± 0.9		3.5 ± 0.9		2.8 ± 0.9		3.0 ± 1.0	
	PhD	3.3 ± 0.6		3.5 ± 0.9		2.9 ± 0.5		3.0 ± 0.7	
Job title	Primary	3.2 ± 0.9	0.13	3.7 ± 0.9	0.06	2.9 ± 0.9	0.51	2.8 ± 1.0	0.021
	Senior	3.3 ± 0.9		3.6 ± 0.8		2.9 ± 0.8		3.1 ± 0.9	
Department	Clinical	3.2 ± 0.9	<0.01	3.7 ± 0.9	0.06	2.9 ± 0.9	<0.01	2.8 ± 0.9	<0.01
	Public health	3.7 ± 0.9		3.5 ± 1.1		2.7 ± 1.0		2.9 ± 1.2	
	Administration	3.7 ± 0.5		3.8 ± 0.7		3.3 ± 0.6		3.3 ± 0.7	
Occupation	Doctor	3.2 ± 0.9	<0.01	3.5 ± 0.9	<0.01	2.8 ± 0.9	<0.01	3.2 ± 0.9	<0.01
	Nurse	3.3 ± 0.9		3.8 ± 0.9		2.9 ± 0.9		2.6 ± 1.0	
	Manager	3.7 ± 0.5		3.8 ± 0.6		3.3 ± 0.6		3.4 ± 0.7	
Working years	≤5	3.2 ± 1.1	0.08	3.6 ± 1.1	<0.01	2.7 ± 1.0	<0.01	2.7 ± 1.1	<0.01
	6–10	3.3 ± 1.0		3.7 ± 1.0		2.9 ± 0.9		2.8 ± 1.0	
	11–15	3.5 ± 0.7		4.0 ± 0.7		3.2 ± 0.6		3.1 ± 0.8	
	>15	3.3 ± 0.6		3.7 ± 0.7		3.1 ± 0.6		3.1 ± 0.7	

**Table 6 ijerph-20-00241-t006:** Binary logistic regression analysis of the symbiotic unit, environment, model, and effect of the integration of medical care and disease prevention.

Variables	Items	Unit OR ^1^ (95% CIs ^2^)	*p*	Environment OR ^1^ (95% CIs ^2^)	*p*	Model OR ^1^ (95% CIs ^2^)	*p*	Effect OR ^1^ (95% CIs ^2^)	*p*
Gender	Male	1.0	0.39	1.0	0.65	1.0	0.67	1.0	<0.01
	Female	1.17 (0.82–1.66)		0.92 (0.65–1.33)		1.07 (0.77–1.48)		1.81 (1.33–2.64)	
Age	≤35	1.0	0.37	1.0	0.18	1.0	0.042	1.0	0.032
	>35	1.15 (0.84–1.57)		1.24 (0.89–1.73)		1.33 (1.01–1.78)		1.36 (1.02–1.83)	
Education	Primary	1.0	0.40	1.0	0.10	1.0	0.66	1.0	0.17
	Advanced	1.17 (0.81–1.71)		1.38 (0.93–2.14)		0.92 (0.65–1.32)		0.78 (0.54–1.12)	
Job title	Primary	1.0	0.26	1.0	0.67	1.0	0.05	1.0	<0.01
	Senior	0.79 (0.53–1.79)		1.10 (0.70–1.12)		0.67 (0.45–1.01)		1.98 (1.31–2.98)	
Department	Clinical	1.0		1.0		1.0		1.0	
	Public health	2.09 (1.12–3.90)	0.02	1.68 (1.04–3.15)	0.04	3.86 (2.15–6.93)	<0.01	1.52 (1.21–1.93)	<0.01
	Administration	1.19 (1.02–1.64)	0.03	1.45 (1.03–2.45)	0.03	1.13 (0.83–1.54)	0.41	0.65 (0.52–0.82)	<0.01
Occupation	Doctor	1.0		1.0		1.0		1.0	
	Nurse	2.02 (1.11–3.66)	0.02	1.33 (0.73–2.43)	0.34	1.47 (0.97–2.22)	0.06	1.51 (0.92–2.51)	0.11
	Manager	1.63 (1.02–2.63)	0.04	0.93 (0.58–1.48)	0.77	3.79 (2.17–6.62)	<0.01	1.63 (1.06–2.49)	0.02
Working years	≤10	1.0	0.87	1.0	<0.01	1.0	<0.01	1.0	0.037
	>10	0.97 (0.71–1.34)		1.67 (1.16–2.40)		1.53 (1.13–2.07)		1.18 (1.07–1.61)	

^1^ OR: odds ratios. ^2^ CIs: confidence intervals.

**Table 7 ijerph-20-00241-t007:** Pearson correlation coefficient analysis.

	Effect	Symbiotic Unit	Symbiotic Environment	Symbiotic Model
Effect	1			
Symbiotic unit	0.799 (*p*< 0.001)	1		
Symbiotic environment	0.813 (*p* < 0.001)	0.882 (*p* < 0.001)	1	
Symbiotic model	0.783 (*p*< 0.001)	0.847 (*p* < 0.001)	0.875 (*p* < 0.001)	1

**Table 8 ijerph-20-00241-t008:** Path explanation and coefficient.

Path Explanation			S.E. ^1^	C.R. ^2^	*p*	Standardized Path Coefficient
Symbiotic unit	→	Symbiosis model	0.054	9.137	<0.001	0.46
Symbiotic environment	→	Symbiosis model	0.058	9.976	<0.001	0.52
Symbiotic environment	↔	Symbiotic unit	0.046	16.021	<0.001	0.91
Symbiosis model	→	Effect	0.093	3.298	<0.001	0.33
Symbiotic unit	→	Effect	0.078	3.676	<0.001	0.29
Symbiotic environment	→	Effect	0.091	3.315	<0.001	0.29

^1^ S.E.: standard error. ^2^ C.R.: critical ratio.

**Table 9 ijerph-20-00241-t009:** Standardized estimates and standard error.

Path Explanation				SE ^1^	Mean	Bias	SE-Bias
a	Symbiotic unit	→	Symbiotic model	0.054	0.462	0.000	0.001
b	Symbiotic model	→	Effect	0.093	0.328	0.000	0.006
c’	Symbiotic unit	→	Effect	0.078	0.287	0.005	0.001

^1^ S.E.: standard error.

**Table 10 ijerph-20-00241-t010:** Results of symbiotic unit, model and effect mediation test.

Effect	Coefficient Term Product		Bootstrapping			
	S.E. ^1^	Mean	Bias-Corrected 95% CIs ^2^		Percentile 95% CIs ^2^	
			Lower	Upper	Lower	Upper
Direct effect (c’)	0.078	0.287	0.112	0.461	0.116	0.468
Direct effect (a)	0.054	0.462	0.323	0.584	0.323	0.583
Direct effect (b)	0.093	0.328	0.103	0.554	0.093	0.548
Mediation effect (ab)	0.152	0.152	0.055	0.271	0.43	0.263
Total effect (c)	0.072	0.439	0.298	0.584	0.296	0.582

^1^ S.E.: standard error. ^2^ CIs: confidence intervals.

## Data Availability

Data are available from the authors at reasonable written request after authorization by the Data Protection Office of the School of Public Health, Hangzhou Normal University, China.

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
