# Peer review of "Influencing Factors and Symbiotic Mechanism of the Integration of Medical Care and Disease Prevention during the COVID-19 Pandemic: A Cross-Sectional Survey of Public Hospital Employees"

_ijerph, 2022, doi:10.3390/ijerph20010241_

Round 1
Reviewer 1 Report
The outbreak of COVID-19 has accelerated the focus on public health in public hospitals, and how to integrate disease prevention and medical care is a management issue. This article uses symbiosis theory to explain the effects of symbiotic units, symbiotic environments and symbiotic models on the disease prevention and medical care functions in public hospitals, which is innovative. The mediating effects of the symbiosis model point to the pathways of action for the effects of disease prevention and healthcare functions in public hospitals, and the findings have high social significance.
Comment 1
Line51-52
The authors describe the reason why public hospitals focus on medical treatment but not on disease prevention is because medical treatment brings more economic effects. But this is primarily an internal issue for hospitals. Do governmental actions and external factors influence public hospitals to value medical care but not disease prevention? I suggest that the authors add to the literature to show the impact of external factors on public hospitals' emphasis on medical care but not disease prevention.
Comment 2
Line 219-220
What is the reason for the author's distinction between employees with ≤10 years of service and those with >10 years of service? Is it the median or the average? I suggest that the authors add the reason for the distinction between years of service.
Comment 3
Line 349-350
The strong correlation between the symbiotic environment and the symbiotic unit has been illustrated in the results of the correlation analysis. I suggest that the authors state here that the symbiotic environment and the symbiotic unit are bidirectional paths.
Comment 4
Line 423-424
The author mentions the lack of human resources for public health as a reason for the difficulties in integrating disease prevention and medical care. However, option B2 describes the capacity of health personnel. I suggest that the authors explain why the lack of health personnel capacity is the reason for the difficulties in integrating disease prevention and health care.
Comment 5
Line 469-470
The hardware of information technology in existing public hospitals is relatively advanced, but the main reason for the barriers to information sharing is a management issue. The aim of establishing an information sharing mechanism is to benefit all employees in public hospitals. I suggest that the authors add details on the establishment of an information sharing mechanism to clarify the positive effects of this objective for public hospitals.
Author Response
Response to Reviewer 1 Comments
The outbreak of COVID-19 has accelerated the focus on public health in public hospitals, and how to integrate disease prevention and medical care is a management issue. This article uses symbiosis theory to explain the effects of symbiotic units, symbiotic environments and symbiotic models on the disease prevention and medical care functions in public hospitals, which is innovative. The mediating effects of the symbiosis model point to the pathways of action for the effects of disease prevention and healthcare functions in public hospitals, and the findings have high social significance.
Thank you for your comments concerning our manuscript entitled “ijerph-2091112 ”, which really helped us improve the manuscript.Those comments are valuable and very helpful. We have read through comments carefully and have made corrections. Based on the suggestion provided in your letter, we uploaded the file of the revised manuscript. Revisions in the text are shown using red highlight for additions. The responses to the reviewer’s comments are marked in red and presented following.
Point 1: Line51-52
The authors describe the reason why public hospitals focus on medical treatment but not on disease prevention is because medical treatment brings more economic effects. But this is primarily an internal issue for hospitals. Do governmental actions and external factors influence public hospitals to value medical care but not disease prevention? I suggest that the authors add to the literature to show the impact of external factors on public hospitals' emphasis on medical care but not disease prevention.
Response 1: We are grateful for the suggestion. In response to your suggestion, we have added to the literature to illustrate the impact of external factors on public hospitals' emphasis on medical treatment over disease prevention. The revised content is as follows:
External factors have also influenced the emphasis on disease prevention in public hospitals. The government's application of finances primarily to the prevention and treatment of public health emergencies has reduced investment in public hospitals leading to further cuts in disease prevention in public hospitals [49]. The government encouraged family doctors to provide public health services to residents in clinics, leading to a reduction in the function of disease prevention in public hospitals [50]. Frequent incidents of food hygiene poisoning in Zhejiang province have caused residents to hesitate about the effectiveness of disease prevention in public hospitals, leading to a decline in the importance of the public health sector [51]. (Line 56-64 in manuscript)
[49] Jin, H., Li, B., Jakovljevic, M. How China controls the Covid-19 epidemic through public health expenditure and policy?. J Med Econ. 2022, 25(1), 437-449. https://doi.org/10.1080/13696998.2022.2054202
[50] Liu, X., Gong, X., Gao, X., Wang, Z., Lu, S., Chen, C., Jin, H., Chen, N., Yang, Y., Cai, M., Shi, J. Impact of Contextual Factors on the Attendance and Role in the Evidence-Based Chronic Disease Prevention Programs Among Primary Care Practitioners in Shanghai, China. Front Public Health. 2022, 9, 666135. https://doi.org/10.3389/fpubh.2021.666135
[51] Chen, L., Wang, J., Zhang, R., Zhang, H., Qi, X., He, Y., Chen, J. An 11-Year Analysis of Bacterial Foodborne Disease Outbreaks in Zhejiang Province, China. Foods. 2022, 11(16), 2382. https://doi.org/10.3390/foods11162382
Point 2: Line 219-220
What is the reason for the author's distinction between employees with ≤10 years of service and those with >10 years of service? Is it the median or the average? I suggest that the authors add the reason for the distinction between years of service.
Response 2: Thank you for your suggestion. The length of employment in public hospitals is differentiated by 10 years. The revised content is as follows:
The rapid development of public health in China has its roots in China's "new health system reform" in 2009, and the impact of the health reform on public hospital employees can be more clearly distinguished by a 10-year distinction [52]. (Line 239-242 in manuscript)
[52] Li, L., Fu, H. China's health care system reform: Progress and prospects. Int J Health Plann Manage. 2017, 32(3), 240–253. https://doi.org/10.1002/hpm.2424
Point 3: Line 349-350
The strong correlation between the symbiotic environment and the symbiotic unit has been illustrated in the results of the correlation analysis. I suggest that the authors state here that the symbiotic environment and the symbiotic unit are bidirectional paths.
Response 3: Thank you for your suggestion. Following your suggestion, We reinterpret the relationship between the symbiotic environment and the symbiotic unit. The revised content is as follows:
The symbiotic environment and symbiotic units interact with each other and behave as bidirectional pathways. (Line 371-373 in manuscript)
Point 4: Line 423-424
The author mentions the lack of human resources for public health as a reason for the difficulties in integrating disease prevention and medical care. However, option B2 describes the capacity of health personnel. I suggest that the authors explain why the lack of health personnel capacity is the reason for the difficulties in integrating disease prevention and health care.
Response 4: Thank you for your suggestion. We explain how the declining capacity of health workers leads to a decline in the effectiveness of disease treatment and prevention. The revised content is as follows:
The reduced capacity of health personnel also leads to a reduction in the effectiveness of disease treatment and prevention. Required competencies for successful chronic disease prevention and health promotion encompass leadership, epidemiology, program practice, and evaluation, among others [53]. However, the training of doctors and nurses in public hospitals lacks these components, leading to a decline in the capacity of health personnel. The lack of time for education and training in disease treatment and prevention due to increased working hours also contributed to this decline in capacity [54]. In addition, doctors and nurses were not able to perform at their normal capacity due to the increased work pressure during the COVID-19 pandemic [55]. (Line 450-459 in manuscript)
[53] Kane, M., Royer-Barnett, J., Alongi, J. Core Competencies for Chronic Disease Prevention Practice. Prev Chronic Dis. 2019, 16, E144. https://doi.org/10.5888/pcd16.190101
[54] Patja, K., Huis In 't Veld, T., Arva, D., Bonello, M., Orhan Pees, R., Soethout, M., van der Esch, M. Health promotion and disease prevention in the education of health professionals: a mapping of European educational programmes from 2019. BMC Med Educ. 2022, 22(1), 778. https://doi.org/10.1186/s12909-022-03826-5
[55] Mao, A., Yang, Y., Meng, Y., Xia, Q., Jin, S., Qiu, W. Understanding the condition of disease prevention and control workforce by disciplines, duties, and work stress during the COVID-19 pandemic: A case from Beijing disease prevention and control system. Front Public Health. 2022, 10, 861712. https://doi.org/10.3389/fpubh.2022.861712
Point 5: Line 469-470
The hardware of information technology in existing public hospitals is relatively advanced, but the main reason for the barriers to information sharing is a management issue. The aim of establishing an information sharing mechanism is to benefit all employees in public hospitals. I suggest that the authors add details on the establishment of an information sharing mechanism to clarify the positive effects of this objective for public hospitals.
Response 5: Thank you for your suggestion. We explain the details of setting up the information sharing mechanism. The revised content is as follows:
The establishment of an information management system in public hospitals breaks down communication barriers and improves management efficiency, based on technical maturity such as ensuring a smooth network. Smart hospital systems can respond quickly to public health emergencies and coordinate the preparation of supplies across departments [56]. Digital storage of disease information improves the efficiency of medical staff and allows data to be quickly shared with relevant departments for early warning [57]. (Line 505-411 in manuscript)
[56] Kuo, Y. W., Tsao, Y. C., Chien, W. C., Huang, Y. M., Liao, L. D. Smart Health Monitoring and Management System for Organizations Using Radio-Frequency Identification (RFID) Technology in Hospitals or Emergency Applications. Emerg Med Int. 2022, 2022, 2177548. https://doi.org/10.1155/2022/2177548
[57] Chen, J., Ruan, Y., Wang, K., Li, L., Yang, Y., Yang, R., Xu, L. Development and Application of a Multidrug-Resistant Tuberculosis Case Management System - Yunnan Province, China, 2017-2020. China CDC Wkly. 2022, 4(38), 855–861. https://doi.org/10.46234/ccdcw2022.177

Reviewer 2 Report
A brief summary
Unlike other countries, public hospitals in China undertake both medical and public health functions, and it is important to rationalise the relationship between medical and public health functions. This article analyses the influence pathways of each element in the integration mechanism of healthcare and disease prevention, as well as the mediating role of the symbiotic model in the symbiotic unit, symbiotic environment and symbiotic effect. The article's conclusions provide a high reference value for the internal management of public hospitals.
Comment 1
In "2. Theoretical hypothesis", the authors use symbiosis theory to explain the symbiotic unit, the symbiotic environment and the symbiosis model in public hospitals. However, the application of symbiosis theory in healthcare is not presented. I suggest that the authors add to the literature in this area and explain the applicability of symbiosis theory to the health care sector.
Comment 2
Line150-151
The first reason why the authors chose public hospitals in Zhejiang Province for their study was that "the investment in public hospitals in Zhejiang Province is among the highest in China". However, this investment is not necessarily in the area of public health within public hospitals. I would suggest that the authors add that they determine whether the investment in public health in Zhejiang Province is also among the highest in China.
Comment 3
In '5.1 Influencing factors of symbiotic unit, environment, model, and effect of the integration of medical and disease prevention', the authors mention that both age and years of experience have a higher perception of the symbiotic model and effect, and that the reasons for this result may be in the same category. I suggest that the authors could present the integration of this factor.
Comment 4
Line 431-435
The authors mention the occurrence of Omicron virus transmission in Zhejiang Province, but do not explain the impact of the outbreak and the response of the symbiotic unit, symbiotic environment and symbiotic model to the event. I suggest that the authors add a description of the role of the symbiotic model in the response to public health emergencies.
Author Response
Response to Reviewer 2 Comments
Unlike other countries, public hospitals in China undertake both medical and public health functions, and it is important to rationalise the relationship between medical and public health functions. This article analyses the influence pathways of each element in the integration mechanism of healthcare and disease prevention, as well as the mediating role of the symbiotic model in the symbiotic unit, symbiotic environment and symbiotic effect. The article's conclusions provide a high reference value for the internal management of public hospitals.
Thank you for your comments concerning our manuscript entitled “ijerph-2091112 ”, which really helped us improve the manuscript.Those comments are valuable and very helpful. We have read through comments carefully and have made corrections. Based on the suggestion provided in your letter, we uploaded the file of the revised manuscript. Revisions in the text are shown using red highlight for additions. The responses to the reviewer’s comments are marked in red and presented following.
Point 1: In "2. Theoretical hypothesis", the authors use symbiosis theory to explain the symbiotic unit, the symbiotic environment and the symbiosis model in public hospitals. However, the application of symbiosis theory in healthcare is not presented. I suggest that the authors add to the literature in this area and explain the applicability of symbiosis theory to the health care sector.
Response 1: We are grateful for the suggestion. Following your suggestion, we have added to the literature to illustrate the applicability of symbiosis theory in health care. The revised content is as follows:
The application of symbiosis theory in health care is mainly in the field of micro human cell health and macro health management. In human cell research, the literature has explored the symbiotic mechanisms between viruses and cells, suggesting that the symbiosis between viruses and normal cells needs to be considered in the treatment process in order to tailor the treatment plan [58-60]. In the field of macro health management, symbiosis theory has been applied to study the synergy between different management departments in the delivery of health services to multiple populations, using various approaches such as information technology to reduce waste of resources and improve management efficiency [61-63]. (Line 100-109 in manuscript)
[58] Suparan, K., Sriwichaiin, S., Chattipakorn, N., Chattipakorn, S. C. Human Blood Bacteriome: Eubiotic and Dysbiotic States in Health and Diseases. Cells. 2022, 11(13), 2015. https://doi.org/10.3390/cells11132015
[59] Batstone, R. T., Lindgren, H., Allsup, C. M., Goralka, L. A., Riley, A. B., Grillo, M. A., Marshall-Colon, A., Heath, K. D. Genome-Wide Association Studies across Environmental and Genetic Contexts Reveal Complex Genetic Architecture of Symbiotic Extended Phenotypes. mBio. 2022, e0182322. Advance online publication. https://doi.org/10.1128/mbio.01823-22
[60] Takeuchi, T., Ohno, H. IgA in human health and diseases: Potential regulator of commensal microbiota. Front Immunol. 2022, 13, 1024330. https://doi.org/10.3389/fimmu.2022.1024330
[61] Ji, W., Qiu, X. Analysis of the Impact of the Development Level of Aerobics Movement on the Public Health of the Whole Population Based on Artificial Intelligence Technology. J Environ Public Health. 2022, 2022, 6748684. https://doi.org/10.1155/2022/6748684
[62] Takase, A., Matoba, Y., Taga, T., Ito, K., Okamura, T. Middle-aged and older people with urgent, unaware, and unmet mental health care needs: Practitioners' viewpoints from outside the formal mental health care system. BMC Health Serv Res. 2022, 22(1), 1400. https://doi.org/10.1186/s12913-022-08838-x
[63] Sreedharan, J. K., Subbarayalu, A. V., AlRabeeah, S. M., Karthika, M., Shevade, M., Al Nasser, M. A., Alqahtani, A. S. Quality assurance in allied healthcare education: A narrative review. Can J Respir Ther. 2022, 58, 103–110. https://doi.org/10.29390/cjrt-2022-009
Point 2: Line150-151
The first reason why the authors chose public hospitals in Zhejiang Province for their study was that "the investment in public hospitals in Zhejiang Province is among the highest in China". However, this investment is not necessarily in the area of public health within public hospitals. I would suggest that the authors add that they determine whether the investment in public health in Zhejiang Province is also among the highest in China.
Response 2: Thank you for your suggestion. We have added information on public health spending in medical institutions in Zhejiang Province over the past five years. The revised content is as follows:
In the last five years, Zhejiang Province has maintained an annual growth rate of 10% in investment in public health in medical institutions, while encouraging a growing number of medical institutions, ranking among the highest in China [64]. (Line 168-170 in manuscript)
[64] Zhejiang Provincial Health and Wellness Commission. Zhejiang Health and Health Statistical Yearbook (2016-2021). Zhejiang Provincial Bureau of Statistics. Printed February 2022 (in Chinese)
Point 3: In '5.1 Influencing factors of symbiotic unit, environment, model, and effect of the integration of medical and disease prevention', the authors mention that both age and years of experience have a higher perception of the symbiotic model and effect, and that the reasons for this result may be in the same category. I suggest that the authors could present the integration of this factor.
Response 3: Thank you for your suggestion. Following your suggestion, We have adjusted the text description of the paragraph. The revised content is as follows:
Age was an influencing factor for symbiotic model and effect. Older public hospital employees had higher perceptions of institutional awareness and effects than did younger employees. The integration of medical and disease prevention was an emer-gency mode after the occurrence of public health events. Older employees in Chinese public hospitals, who mostly experienced two public health events, severe acute res-piratory syndrome (SARS) and COVID-19, were more affected by such events than did younger employees, leading to higher perceptions of model and effect [27]. Related to this result, employees with longer working years had higher perceptions of the symbiotic environment, model, and effects. After a public health emergency in China, employees of public hospitals with higher qualifica-tions were often required to directly participate in health emergency work, so their perception was higher than did younger employees [34].Public health department employees had higher perceptions of the symbiotic unit, environment, model, and the effect of the integration of medical and disease prevention than did clinical department employees and administrative employees. The main reason is that starting from 2021, the Zhejiang provincial government required that medical institu-tions above the second level must set up a public health department. The public health department received government funding in health resources and systematically trained public health department employees [28]. Since administrators were responsible for public health training in public hospitals, their knowledge of symbiotic unit, envi-ronment, and models were also higher than did clinicians [29]. However, employees of clinical departments have been engaged in medical technology work for a long time, and their work content does not include public health knowledge, which lead to their lack of awareness of the integration of medical and disease prevention [30]. It is important to increase awareness by conducting training on the integration of medical and disease prevention for clinical department employees. Unlike previous studies which found that doctors were more aware of public health [31-32], this study found that nurses were more aware of symbiotic units than did doctors. The reason was that in China, the oc-currence of nosocomial infections had a greater impact on the careers of nurses, which made nurses more sensitive to health emergencies [33]. (Line 412-440 in manuscript)
Point 4: Line 431-435
The authors mention the occurrence of Omicron virus transmission in Zhejiang Province, but do not explain the impact of the outbreak and the response of the symbiotic unit, symbiotic environment and symbiotic model to the event. I suggest that the authors add a description of the role of the symbiotic model in the response to public health emergencies.
Response 4: Thank you for your suggestion. We explain the role of symbiotic patterns in the COVID-19 pandemic. The revised content is as follows:
Healthcare facilities with a symbiotic model demonstrated higher treatment efficiency and faster response times in response to the COVID-19 pandemic. This has directly reduced the health risks to the population of Zhejiang Province. (Line 467-470 in manuscript)
